# Modeling of the Electrochemical Motion Sensor Conversion Factor at High Frequencies

**DOI:** 10.3390/mi13020153

**Published:** 2022-01-20

**Authors:** Vadim Agafonov, Iuliia Kompaniets, Bowen Liu, Jian Chen

**Affiliations:** 1Moscow Institute of Physics and Technology, 141701 Dolgoprudny, Russia; kompaniets.yu@ya.ru; 2State Key Laboratory of Transducer Technology, Aerospace Information Research Institute, Chinese Academy of Sciences, Beijing 100190, China; liubowen17@mails.ucas.ac.cn (B.L.); chenjian@mail.ie.ac.cn (J.C.)

**Keywords:** electrochemical sensor, MET sensor, microelectrodes, microhydrodynamics, diffusion, electrolyte, sensitivity, conversion factor

## Abstract

The conversion factor of the electrochemical motion sensors at low frequencies is usually quite high. At the same time, it decreases significantly with the increase in frequency. Thus, increasing the conversion factor for high frequencies is essential for practical use. In this work, the theoretical model that allows establishing the basic laws governing the conversion of high-frequency signals in an electrochemical cell has been suggested. The approach was based on the fact that in the case of high frequencies, the diffusion length is less than the distance between the electrodes and the thickness of the channel and it is enough to consider the transformation of the fluid motion into electrical current only near the cathodes. It was found that the signal output current can be represented as the sum of the term which is proportional to the steady-state concentration gradient along the surface on which the cathode is located, and the term proportional to the concentration gradient normal to the surface. Both first and second terms and the total signal current have been calculated for a particular case of a four-electrode planar system. The practical conclusion is that the high frequency conversion factor increases with the interelectrode distance and the channel width decreases compared to the cathode dimension.

## 1. Introduction

Electrochemical cells with characteristic geometrical dimensions of electrode systems from one to several hundred micrometers are used as sensitive elements of motion sensors and wave fields in seismometers, accelerometers, geophones and hydrophones [1,2,3,4,5,6,7,8,9,10,11,12,13]. The most important advantage of electrochemical sensors is a high conversion coefficient in combination with a rather simple design, the production of which on a mass scale is possible with the use of modern micromachined technologies. The fields of application of such sensors, such as seismology, seismic exploration and structural monitoring require measuring weak signals. Therefore, many works published in recent years have been devoted to the study of ways to increase the conversion factor of sensors of this type [14,15,16,17].

Increasing the conversion factor for high frequencies is of particular importance for practical use. The fact is that at low frequencies, the conversion coefficient of electrochemical sensors is always quite high. Even the very first sensors of this type had extremely high sensitivity at low frequencies [18,19]. However, with an increase in frequency, the conversion coefficient of early versions of electrochemical sensors decreased dramatically, which limited the range of the first samples to a band of up to 0.5 Hz [18,19,20,21,22]. Later, thanks to the efforts of many researchers and technological advances, the sensitivity at high frequencies and the measurement range were significantly increased. Nevertheless, when compared with other technologies (piezoelectric, electrodynamic sensors), a typical situation is when electrochemical sensors are many times superior in sensitivity to their counterparts at low frequencies and begin to yield to them as the frequency increases.

It is difficult to optimize the converting system for high frequencies because the signal conversion process depends on the microscopic structure of the converting system, in particular, on the shape of the electrodes of the electrochemical cell and on the peculiarities of the placement of the electrodes relative to the surrounding dielectric elements of the structure of the electrochemical cell. Therefore, theoretical models should take into account the micro-scale geometry of the electrodes, and experimental samples should be manufactured with high accuracy.

If a special identification of approximations is not used, the calculations require a detailed description of the distribution of velocities and concentrations at small distances from the electrodes surface and require the division of the studied area into smaller parts with high detail, which increases the amount of computation and makes numerical modeling of characteristics at high frequencies quite difficult.

As a result, most of the previously performed calculations have been performed for a limited area of high frequencies [22,23]. Until recently, the calculations were carried out mainly for frequencies not exceeding 10–50 Hz. Recent calculations have been carried out for frequencies up to 100 Hz [24]. The same methods as for lower frequencies have been used, while the objectives of the study were not set to determine the nature of regularities specific for high frequencies according to the conversion coefficient depending on the geometry of the transforming cell, related specifically to the high-frequency area.

The aim of this work was to create a theoretical model that allows the establishment of the basic laws governing the conversion of high-frequency signals in an electrochemical cell. The object of the study was a planar converting cell. In the case of high frequencies, when the diffusion length becomes less than the distance between the electrodes and the size of the channel where the working liquid flows, it is sufficient to consider the transformation of the fluid motion only near one of the electrodes (cathode). The role of other electrodes is reduced to the creation of a stationary concentration field, the interaction of which with hydrodynamic flows causes the output signal in electrochemical sensors. It was found that the signal output current can be represented as the sum of two terms. The first term is proportional to the steady-state concentration gradient along the surface on which the electrode is located, the second term depends on the concentration gradient normal to the surface. Generally speaking, the contributions of these terms to the signal current are comparable to each other, however, the quantitative ratio is determined by the stationary concentration distribution, and consequently by the design of the entire electrode system. For a particular case of a four-electrode planar system of electrodes located on the walls of a channel with flat walls, the stationary concentration field was calculated and the dependences of each of the terms and the total signal current on the distance between the electrodes, their dimensions and the thickness of the channel in which the working fluid flows were established.

## 2. Materials and Methods

The electrochemical cell, used as a transforming element of a motion or wavefield sensor, in which the working fluid is an aqueous solution of potassium or lithium iodide with a small addition of molecular iodine, is the most common option. Molecular iodine in the solution is in the form of triiodide ions $I_{3}^{-}$. The following electrochemical reaction takes place on the electrodes [25,26]:$$I^{-}⇆I_{3}^{-}+2e\;$$

In each of these elementary reactions, two electrons are transferred across the electrode surface. Thus, the electric current can be calculated if the flux of tri-iodide ions through the electrode surface is determined. This fact is used in a model based on convective diffusion, which makes it possible to exclude other types of ions from the analysis and to consider the transport of only tri-iodide ions, which simplifies the mathematical problem. Tri-iodide ions are called the active component of the solution. In principle, the subsequent analysis is applicable to other redox systems as well.

The electrochemical cell is a microscopic structure containing many channels filled with a working fluid. In the channels, the fluid flow caused by an external mechanical signal flows in the vicinity of the electrodes, between which a constant potential difference is applied. As a rule, the arrangement of the electrodes along the fluid flow corresponds to the anode-cathode-cathode-anode scheme (ACCA layout). For many designs developed, the electrodes are placed on the walls of the channels, and their sizes satisfy the relation L≫b≫w, where L, b are the sizes of the electrodes along the surface of the channel wall perpendicular and parallel to the fluid flow, respectively, and w is the electrode thickness. Such cells are called planar [27,28] and they are looked into in this paper.

### 2.1. Mathematical Formulation of the Problem

The mathematical model combines the following equations.

(1)Navier–Stokes equation for incompressible liquid:


(1)
$$\frac{\partial \vec{v}}{\partial t}=\nu \Delta \vec{v}-\frac{1}{\rho }\nabla p,$$


(2)the continuity equation for incompressible liquid:


(2)
$$div\;\vec{v}=0,$$


(3)the convective diffusion equation for the active component of the electrolyte:

(3)$$\frac{\partial c}{\partial t}=D\Delta c-\left(\vec{v}\nabla \right)c,$$
where v is the velocity of the electrolyte solution in the channel, ν is the kinematic viscosity of the electrolyte solution, ρ is the density of the electrolyte solution, D is the diffusion coefficient of the active component of the electrolyte solution, c is the concentration of the active component of the electrolyte solution.

If the distribution of the concentration of the active ions is known, the electric current through the electrode surface can be calculated according to the following expression:(4)$$I=2Dq\oint_{S_{e}}^{}\left(\nabla c,n\right)\;dS,$$

Here the integration is over the electrode surface, q is the electron charge, n is the normal vector to the surface. Coefficient 2 takes into account that two electrons are involved in one electrochemical reaction on the surface. As a rule, the current flowing through the cathode is used as a signal.

At low speeds of the electrolyte solution, the concentration of the active component can be presented as an expansion in powers of speed $\vec{v}$:(5)$$c=c_{st}+c_{1},$$
where $c_{st}$ is the concentration of the active component in the stagnant electrolyte solution, $c_{1}$ is the linear in speed addition to the concentration. Substituting the Equation (5) into (3) and discarding the terms of the second and higher order of smallness, obtain:(6)$$\Delta c_{st}=0,$$
(7)$$\frac{\partial c_{1}}{\partial t}-D\Delta c_{1}=-\left(\vec{v}\nabla \right)c_{\mathrm{st}},$$

The boundary conditions for hydrodynamic Equations (1) and (2) are no-slip conditions on the channel walls. For Equation (3), a simple boundary condition of zero concentration for the cathodes and the constant concentration on the anodes are used similar to many previously published studies [21,29,30,31]. Besides, the dielectric boundaries will be considered as non-penetrable for active ions.

The boundary conditions for the concentration $c_{st},\;c_{1}$ can be formulated as follows:(8)$${c_{st}|}_{x\in S_{c}}=0,\;{c_{st}|}_{x\in S_{a}}=c_{a},\;{c_{1}|}_{x\in S_{c},S_{a}}=0,\;{\left(\nabla c_{1},n\right)|}_{x\notin S_{c},S_{a}}={\left(\nabla c_{1},n\right)|}_{x\notin S_{c},S_{a}}=0,$$
where $S_{c},S_{a}\;$are the surfaces of cathodes and anodes, respectively. The first three equations express the condition of zero concentration for the cathodes and the constant concentration on the anodes, the last equation expresses the absence of ion flux through the dielectric surface.

To study the characteristics of a planar transforming element, we can use two-dimensional equations.

### 2.2. Non-Stationary Convective Diffusion

A distinctive feature of the conversion of a mechanical signal into an electric current at high frequencies ω is the smallness of the diffusion length $r_{D}=\sqrt{D/\omega }$ in comparison with the distance between the electrodes and the channel width. Therefore, when solving the nonstationary diffusion Equation (7), it is sufficient to consider each electrode independently of the others. The role of other electrodes is to create a steady-state concentration $c_{st}$ presented on the right side of Equation (7).

Assume the stationary concentration distribution and the hydrodynamic velocity to be known and find a solution to Equation (7) near the cathode. The cathode is a long thin strip located on a flat non-conducting surface. Consider the fluid flow to be directed perpendicular to the long side of the electrode (Figure 1).

In Equation (7), carry out the Fourier transform in time and coordinate x:(9)$$\frac{\partial^{2}c_{1}}{\partial z^{2}}-\left(k^{2}+\frac{i\omega }{D}\right)c_{1}=L\left(k,z\right),$$

Here L(k,z) is given by the following expression:(10)$$L\left(k,z\right)=\int_{-\infty }^{+\infty }L\left(x,z\right)e^{-ikx}dx,\;L\left(x,z\right)=\frac{1}{D}\left(v_{x}\frac{\partial c_{st}}{\partial x}+v_{z}\frac{\partial c_{st}}{\partial z}\right),$$

Additionally, take into account that at high frequencies the volume near the channel surface, i.e., for small z, is of high interest and use the following expansion for the stationary concentration $c_{st}$:(11)$$c_{st}\left(x,z\right)=c_{st}\left(x,0\right)+\frac{\partial c_{st}}{\partial z}\left(x,0\right)z,$$

The importance of preserving not only the first but also the second term of the expansion will be clear from what follows. Additionally, take into account that the tangential velocity decreases when approaching the surface according to a linear law, while the normal component decreases according to a quadratic law, and represent the fluid velocity near the surface in the following form:(12)$$v_{x}\left(x,z\right)=\tau \left(x\right)z,\;v_{z}\left(x,z\right)=n\left(x\right)z^{2},$$

Accordingly, take into account (10)–(12) and obtain:(13)$$L\left(x,z\right)=\frac{z}{D}\left[\tau \left(x\right)\frac{\partial c_{st}\left(x,0\right)}{\partial x}+z\left(\tau \left(x\right)\frac{\partial^{2}c_{st}\left(x,0\right)}{\partial z\partial x}+n\left(x\right)\frac{\partial c_{st}\left(x,0\right)}{\partial z}\right)\right],$$

The solution to Equation (9) can be represented as follows:(14)$$c_{1}\left(k,z\right)=\frac{j_{1}\left(k\right)}{2Dq\lambda }e^{-\lambda z}+\frac{1}{\lambda }\int_{0}^{z}L\left(k,\zeta \right)sinh(\lambda z-\lambda \zeta )d\zeta -\frac{cosh\lambda z}{\lambda }\int_{0}^{d}L\left(k,\zeta \right)e^{\lambda d-\lambda \zeta }d\zeta ,$$

Here, λ2=k2−iωD, Reλ>0.$j_{1}\left(k\right)=\frac{1}{2\pi }\int_{\xi \in S_{c,},S_{a,}}j_{1}\left(\xi \right)e^{-ik\xi }d\xi$ is the Fourier transform of the electric current density on the surface.

Take z=0 and carry out the inverse Fourier transform, assuming, according to the boundary conditions (8), that the concentration on the surface of the electrodes $c_{1}=0$:(15)$$\int_{-\infty }^{\infty }\frac{dk}{\lambda }\int_{s_{c}}j_{1}\left(\xi \right)e^{-ik\xi +ikx}d\xi =2Dq\int_{0}^{\infty }d\zeta \int_{-\infty }^{+\infty }L\left(\xi ,\zeta \right)d\xi \int_{-\infty }^{+\infty }\frac{e^{-\lambda \zeta +ik\left(x-\xi \right)}}{\lambda }dk,$$

In this equation, the x coordinate corresponds to a point on the surface of the electrode. Substitute (14) and carry out the integration over ζ on the right-hand side of the equation, and replace $\lambda \to \lambda_{D}$ on the left-hand side of the equation for high frequencies (here $\lambda_{D}=\sqrt{\frac{i\omega }{D}}$) to obtain the following expression:(16)$$j_{1}\left(x\right)=2q\lambda_{D}\int_{-\infty }^{+\infty }d\xi \int_{-\infty }^{+\infty }\left(\tau \left(\xi \right)\frac{\partial c_{st}\left(\xi ,0\right)}{\partial \xi }\frac{1}{\lambda^{3}}+\left(\tau \left(\xi \right)\frac{\partial^{2}c_{st}\left(\xi ,0\right)}{\partial z\partial x}+n\left(\xi \right)\frac{\partial c_{st}\left(\xi ,0\right)}{\partial z}\right)\frac{2}{\lambda^{4}}\right)e^{ik\left(x-\xi \right)}dk,$$

For the term containing $\frac{\partial^{2}c_{st}\left(x,0\right)}{\partial z\partial x}$, integrate over ξ by parts:(17)$$j_{1}\left(x\right)=2q\lambda_{D}\int_{-\infty }^{+\infty }d\xi \int_{-\infty }^{+\infty }\left[\tau \left(\xi \right)\frac{\partial c_{st}\left(\xi ,0\right)}{\partial \xi }\frac{1}{\lambda^{3}}+\left(\tau \left(\xi \right)ik+n\left(\xi \right)\right)\frac{2}{\lambda^{4}}\frac{\partial c_{st}\left(\xi ,0\right)}{\partial z}\right]e^{ik\left(x-\xi \right)}dk,$$

It can be seen from the obtained expression that the regions in ξ that make a nonzero contribution to the integral are different for the first and second terms in the expression in square brackets. The first term is nonzero only outside the electrode since the stationary concentration on the cathode surface is constant due to the boundary conditions. On the contrary, the second term is nonzero only at the electrode, due to the boundary condition, which expresses the impossibility of current flow through the dielectric surface. Accordingly, expression (17) can be represented as follows:(18)$$j_{1}\left(x\right)=2q\lambda_{D}\left[\int_{\xi \notin S_{c}}^{}\int_{-\infty }^{+\infty }\tau \left(\xi \right)\frac{\partial c_{st}\left(\xi ,0\right)}{\partial \xi }\frac{1}{\lambda^{3}}e^{ik\left(x-\xi \right)}dkd\xi +\int_{\xi \in S_{c}}^{}\int_{-\infty }^{+\infty }\left(\tau \left(\xi \right)ik+n\left(\xi \right)\right)\frac{2}{\lambda^{4}}\frac{\partial c_{st}\left(\xi ,0\right)}{\partial z}e^{ik\left(x-\xi \right)}dkd\xi \right],$$

Returning to the expansion (11), note that it is precisely this circumstance that makes it necessary to preserve the term $\sim z^{2}$. On the one hand, in the first term in (18), the parameter 1/λ appears to a lesser extent than in the first term, which should correspond to a weaker drop with increasing frequency. On the other hand, the integration for the first term is carried out over the region outside the electrodes corresponding to large absolute values (x−ξ). As a result, it becomes impossible to determine which of the terms in expression (18) are dominant before performing the calculations.

Compared to the methods based on the numerical solution of the Equation of nonstationary convective diffusion (7), Equation (18) for calculating the high-frequency response of an electrochemical transforming element has the advantage that it does not require computer time-consuming and detailed concentration distribution $c_{1}$ in the near-electrode area, which is demanding on the accuracy of calculation results and is essential for the calculation of the gradient according to (4), and the problem is reduced to a simple integration of the parameters characterizing the stationary concentration distribution.

The final result of the calculation is the electric current flowing through the cathodes under the action of the hydrodynamic motion of the liquid, which can be found by integrating Equation (18) over the electrode surface.

### 2.3. Hydrodynamics

To perform subsequent calculations, specify the geometry of the transforming element. Assume all the electrodes to be located on one wall of the channel bounded by two infinite planes. b is the size of the electrodes, $a_{c}$ is the distance between the cathodes, a is the distance between the anode and the adjacent cathode, d is the thickness of the channel. The cell is assumed to be symmetric, and the origin of the coordinate system is placed on the channel wall containing the electrodes, at the central point of the electrode structure, as shown in Figure 2.

Suppose an external mechanical signal causes the fluid to flow through the channel with the volumetric flow Q and frequency ω. Taking into account the boundary conditions, obtain the expression for the x-component of the velocity of the electrolyte solution in the channel:(19)vx(z)=QdLsinh(αd)−sinh(αz)−sinh(αd−αz)sinh(αd)−2αd(cosh(αd)−1),

Here $\alpha =\sqrt{\frac{i\omega }{\nu }}$. At small distances from the surface, the expression for the velocity can be represented by a term linear in z.
(20)$$v_{x}\left(z\right)=\tau z,$$

Here $\tau =\frac{6Q}{d^{2}L}f\left(\alpha d\right)$. In the limiting cases of high and low hydrodynamic frequencies:(21)f(αd)=αd,  |αd|≫1f(αd)=1,  |αd|≪1

### 2.4. Distribution of the Stationary Concentration

Calculate the stationary concentration distribution. Similarly to [31], use the Fourier transform method. After the Fourier transform by x coordinate in (6), go to the Fourier component $j_{0}\left(k\right)\;$of the electrical current density $j_{0}\left(x\right)=-2Dq{\frac{\partial C_{st}}{\partial z}|}_{z=0}$ the concentration distribution can be calculated from the equation:(22)$$c_{st}\left(k,z\right)=\frac{\cosh k\left(d-z\right)}{\sinh kd}\frac{j_{0}\left(k\right)}{2Dqk}+c_{0},$$

Taking into account the boundary conditions (7) using the relation $j_{0}\left(\xi \right)=j_{0}\left(-\xi \right)$ and transforming to dimensionless values: $\tilde{a},\;\tilde{d,\;}\;\tilde{a_{c}}=a,d,\;a_{c}$/*b*.$\tilde{j}=\frac{jb}{2\pi Dqc_{0}},$ $\tilde{c}=\frac{c}{c_{0}}$ obtain the following integral equation to determine the unknown function $\tilde{j_{0}}\left(\tilde{\xi }\right)$:(23)$$\int_{0}^{+\infty }d\tilde{k}\int_{S_{el}}\frac{\cosh\tilde{k}\tilde{d}}{\sinh\tilde{k}\tilde{d}}\frac{\tilde{j_{0}}\left(\tilde{\xi }\right)}{\tilde{k}}\cos\tilde{k}\tilde{x}\cos\tilde{k}\tilde{\xi }d\tilde{\xi }=\{\begin{array}{c} -1,x\in \left[\frac{{\tilde{a}}_{c}}{2},\frac{{\tilde{a}}_{c}}{2}+1\right]\\ {\tilde{c}}_{a},x\in \left[\frac{{\tilde{a}}_{c}}{2}+\tilde{a}+1,\frac{{\tilde{a}}_{c}}{2}+\tilde{a}+2\tilde{b}\right] \end{array},$$
${\tilde{c}}_{a}$ is a priori unknown additive, the method for determining which is described below.

Split the anode and cathode into N equal sections each. Within each section, the electric current density will be considered constant.

Additionally, take into account that the currents at the anode and cathode must be equal to each other, as required from the law of conservation of charge: (24)$$\tilde{j_{0}}=-i_{0}+\sum_{n=1}^{N}A_{n}(\Theta \left(\tilde{\xi }-{\tilde{x}}_{n}-\frac{1}{2N}\right)-\Theta \left(\tilde{\xi }-{\tilde{x}}_{n}+\frac{1}{2N}\right)),\;{\tilde{x}}_{n}=\frac{\tilde{a_{c}}}{2}+\frac{n}{N}-\frac{1}{2N},\;\tilde{\xi }\in \left[\frac{\tilde{a_{c}}}{2},\frac{\tilde{a_{c}}}{2}+1\right],\;\tilde{j_{0}}=i_{0}+\sum_{n=1}^{N}B_{n}(\Theta \left(\tilde{\xi }-{\tilde{y}}_{n}-\frac{1}{2N}\right)-\Theta \left(\tilde{\xi }-{\tilde{y}}_{n}+\frac{1}{2N}\right)),\;\;\;\;\;\;\;\;\;{\tilde{\;y}}_{n}=\tilde{a}+1+\frac{\tilde{a_{c}}}{2}+\frac{n}{N}-\frac{1}{2N},\;\tilde{\xi }\in \left[\frac{\tilde{a_{c}}}{2}+\tilde{a}+1,\frac{\tilde{a_{c}}}{2}+\tilde{a}+2\right],$$

$i_{0}$ is the average electric current density on the electrode surface. $A_{n},\;B_{n}$ are the unknown coefficients, the sum of which is equal to 0, i.e., they satisfy additional conditions: $A_{N}=-\overset{n=N-1}{\underset{n=1}{\sum }}A_{n};B_{N}=-\sum_{n=1}^{n=N-1}B_{n}.\;$

Introduce the notation $a_{n}=\frac{\pi A_{n}}{4i_{0}},\;b_{n}=\frac{\pi B_{n}}{4i_{0}}$, $c_{0}^{*}=\frac{\pi \tilde{c_{0}}}{4i_{0}},\;c_{a}^{*}=\frac{\pi \tilde{c_{a}}}{4i_{0}}.$ After substitution of expressions (24) in (23), integration and transformation to matrix form, obtain the system: (25)$$\left(a_{n},i_{0},b_{n},c_{a}\right)M=\left(\left(-1..-1\right),\left(0..0\right)\right),$$
where M is a square matrix of size 2N×2N:(26)$$M=\left[\begin{array}{cc} A_{cc} & A_{ca}\\ \left\|-X\right\| & \left\|Y\right\|\\ A_{cc} & A_{aa}\\ \left(0..0\right) & \left(-1..-1\right) \end{array}\right],$$

$A_{cc},\;A_{ac},A_{ca},A_{aa}\;$ are the matrices (N−1)×N, elements of which are specified by the expression:(27)$$A_{cc,nl}=\int_{0}^{\propto }\frac{coshkd}{k^{2}sinhkd}sin\frac{k}{2N}sin\frac{k}{2}\left(x_{N}-x_{n}\right)sin\frac{k}{2}\left(x_{N}+x_{n}\right)coskx_{l}dk,\;A_{ca,nl}=\int_{0}^{\propto }\frac{coshkd}{k^{2}sinhkd}sin\frac{k}{2N}sin\frac{k}{2}\left(y_{N}-y_{n}\right)sin\frac{k}{2}\left(y_{N}+y_{n}\right)coskx_{l}dk,\;A_{ac,nl}=\int_{0}^{\propto }\frac{coshkd}{k^{2}sinhkd}sin\frac{k}{2N}sin\frac{k}{2}\left(x_{N}-x_{n}\right)sin\frac{k}{2}\left(x_{N}+x_{n}\right)cosky_{l}dk,\;A_{aa,nl}=\int_{0}^{\propto }\frac{coshkd}{k^{2}sinhkd}sin\frac{k}{2N}sin\frac{k}{2}\left(y_{N}-y_{n}\right)sin\frac{k}{2}\left(y_{N}+y_{n}\right)cosky_{l}dk,$$

l=1..N, n=1..N−1. (0..0), (−1..−1) are the rows of N length comprising of “0” and “−1” correspondingly. ‖X‖ and ‖Y‖ are the strings of length N, components of which are calculated according to the following expressions (*l* = 1…*N*)):(28)$$X_{l}=-\int_{0}^{\infty }\frac{coshkd}{k^{2}sinhkd}sin\frac{k}{2}sink\frac{1+a}{2}sink\left(1+\frac{a+a_{c}}{2}\right)coskx_{l}dk,\;Y_{l}=\int_{0}^{\infty }\frac{coshkd}{k^{2}sinhkd}sin\frac{k}{2}sink\frac{1+a}{2}sink\left(1+\frac{a+a_{c}}{2}\right)cosky_{l}d,$$

l=1…N. After the solution of the system of Equation (26) has been found, Equations (22) and (24) allow calculating the concentration distribution in space and the electric current density on the surface of the electrodes.

For the subsequent calculation, see Equation (18) in dimensionless form:(29)$${\tilde{j}}_{1}\left(\tilde{x}\right)=\frac{\tilde{Q}{\tilde{\lambda }}_{D}}{{\tilde{d}}^{2}}\left[-\int_{\tilde{\xi }\in S_{c}}^{}\int_{0}^{+\infty }\frac{\partial {\tilde{c}}_{st}\left(\tilde{\xi },0\right)}{\partial \tilde{z}}\frac{2\tilde{k}}{{\tilde{\lambda }}^{4}}sin\tilde{k}\left(\tilde{x}-\tilde{\xi }\right)d\tilde{k}d\tilde{\xi }+\int_{\tilde{\xi }\notin S_{c}}^{}\int_{-\infty }^{+\infty }\frac{\partial {\tilde{c}}_{st}\left(\tilde{\xi },0\right)}{\partial \tilde{\xi }}\frac{1}{{\tilde{\lambda }}^{3}}cos\tilde{k}\left(\tilde{x}-\tilde{\xi }\right)d\tilde{k}d\tilde{\xi }\right],$$

Here, the following notation is used: ${\tilde{\lambda }}_{D}=\sqrt{i\tilde{\omega }}=\sqrt{\frac{i\omega b^{2}}{D},\;}\tilde{Q}=\frac{6Q}{\pi LD}f(\alpha d)$.

## 3. Results

The resulting equations were used to calculate the conversion factor for the following ranges of geometric characteristics of the converting electrochemical cell: $0.5<\;\tilde{d}<10;\;0.5<\;\tilde{a}<10;\;0.5<{\;\tilde{a}}_{c}<10.$ For each set of characteristics, the stationary problem was first solved and the electric current density on the electrode surface was determined from Equation (24). Then, from (22), the stationary concentration distributions were calculated and the distribution of the tangential and normal components of the concentration gradient on the channel wall was determined. Finally, Equation (29) was used to calculate the signal current.

### 3.1. Stationary Concentration and Gradient of the Stationary Concentration

An example of the distribution of the stationary electrode current density ${\tilde{j}}_{0}$ according to Equation (24) is shown in Figure 3. In this example, the following geometric parameters of the system are used: $\tilde{d}=2,\;\tilde{a}=1,\;{\tilde{a}}_{c}=1$. The obtained distribution is characterized by the presence of pronounced maximums of the current density at the edges of the electrodes because of the diffusion of active carriers on these parts of the electrodes not only from the area directly above the electrode but also from the areas adjacent to the electrodes.

The spatial distribution of the stationary concentration at the same parameters of the electrochemical cell is shown in Figure 4 for the cell volume, and in Figure 5 for the channel wall. In particular, it can be seen from Figure 5 that the obtained concentration distribution does correspond to the given boundary conditions. Besides, the obtained dependencies qualitatively correspond to the results of calculations performed in [31], which confirm the equivalence of the calculation methods and can serve as a method for verifying a computer code.

Equation (29) implies that the signal current is expressed through the distribution of the normal and tangential components of the steady-state concentration gradient. For several sets of geometries, the corresponding plots are shown in Figure 6 for tangential components and in Figure 7 for normal components.

From these figures, for all parameters of the system, the components of the concentration gradient have pronounced extrema at the edges of the electrodes. The indicated peak values deserve special attention since these peaks make the most significant contribution to the integrals of expression (29) when calculating the signal current. Table 1 shows the values of the absolute peak values of the components of the concentration gradient near the cathode edges. The calculations were carried out for two values of the number of segments into which the electrode N is divided: N=40 and N=80 and the difference in the results obtained is presented as an error.

The presented data show that with a decrease in the channel thickness $\tilde{\;d}$, the concentration gradient on both sides of the cathode decreases, a decrease in $\tilde{a}\;$leads to an increase in the gradients, and the value of ${\tilde{a}}_{c}$ has very little effect on the values of the concentration components on the cathode side facing the anode, but decreases their values at the opposite edge of the cathode. In this case, a decrease in any of the sizes $\tilde{d},\tilde{a,}\;{\tilde{a}}_{c}$ increases the difference between the concentration gradients on the opposite sides of the cathode.

### 3.2. Signal Current Dependence on the Geometrical Characteristics

After the distributions of the stationary concentration of the active component were determined, the stationary and signal cathode currents were calculated depending on the geometric characteristics of the converting system. The signal current was determined at $\tilde{\omega }=10.$ Figure 8 shows the dependence of the stationary background current (Figure 8a) and the signal current (Figure 8b) on the channel width $\tilde{d}$, while Figure 9 and Figure 10 show similar dependencies on the interelectrode distances $\tilde{a,}\;{\tilde{a}}_{c}$.

The obtained results show that the stationary and signal currents depend on the geometry of the electrochemical cell in different ways. The stationary current increases with an increase in the channel thickness (Figure 8) and the intercathode distance (Figure 10) and it decreases with an increase in the anode-cathode distance (Figure 9). The signal current generally increases with a decrease in any of the studied geometric parameters. The only exclusion is in the dependence of the signal current on the distance between anode and cathode, which has a minimal value at ${\hat{a}}_{c}\;\sim \;0.5$, slightly increases at a large distance and grows noticeably at a small distance.

Moreover, the frequency response of the output signal was investigated in the frequency range of dimensionless frequencies from 10 to 1000. The condition |αd|≫1 is considered to be valid. The graph presented in Figure 11 shows that the frequency dependence has a character of ~1f1.4, which practically coincides with the dependence of ~1f1.5, as was found experimentally in a number of works [32,33,34,35].

## 4. Discussion

In this work, high frequencies are understood as signal frequencies at which the diffusion length is significantly less than the distance between the electrodes. For such frequencies, the conversion of signals at each electrode occurs independently of the others. Even in this case, the relative position of the other electrodes fundamentally affects the conversion coefficient. Mathematically it is seen as the expression for the output current that contains two terms proportional to the components of the steady-state concentration gradient $\frac{\partial {\tilde{c}}_{st}\left(\tilde{\xi },0\right)}{\partial \tilde{z}}$ and $\frac{\partial {\tilde{c}}_{st}\left(\tilde{\xi },0\right)}{\partial \tilde{\xi }}\;.\;$The values of these quantities essentially depend on the position of the other electrodes and the size of the channel in which the liquid flows. The calculations performed for a cell, which is a system of flat electrodes deposited on the channel walls, show that the conversion coefficient increases with a decrease in the distance between the electrodes, as well as with a decrease in the channel thickness. The indicated dependencies correlate well with the previously obtained experimental and calculated data. Experimentally, an increase in the conversion coefficient at high frequencies with a decrease in the cathode-anode distance was observed in [28,32,33], with a decrease in the intercathode distance in [31]. According to [33], the conversion coefficient increases with decreasing thickness of the working channel with other parameters being the same, also in agreement with the results obtained in this work. In the latter case, however, it should be borne in mind that, in contrast to the geometry considered here, in [33] only cathodes were located on the channel walls, and the anodes were placed outside the channel. It is important to note that there is no direct correlation between the behavior of the background and signal currents. For example, the background current always increases with increasing channel thickness, while the signal current has an opposite trend. At the same time, there is a clear correspondence between the changes in the conversion coefficient and the difference between the absolute values of the gradient of stationary concentration $\frac{\partial {\tilde{c}}_{st}\left(\tilde{\xi },0\right)}{\partial \tilde{z}}$ and $\frac{\partial {\tilde{c}}_{st}\left(\tilde{\xi },0\right)}{\partial \tilde{\xi }}\;$at the cathode edges. An increase in this difference leads to an increase in the conversion factor, regardless of which cell parameters have been changed.

The conclusion here is that an increase in the conversion coefficient can be achieved by increasing the size of the electrodes themselves. In this case, because of the decrease in the influence of the adjacent anode on the concentration distribution at the opposite end of the cathode, the difference between the values of the steady-state concentration gradient at the opposite ends of the cathode will also increase.

Thus, the practical result of the calculations performed are ways to increase the sensitivity of the electrochemical sensor at high frequencies. For the case of a flat channel, on the walls of which the electrodes are located, the conversion factor can be increased by decreasing the distance between the electrodes and the thickness of the channel, as well as increasing the size of the electrodes.

The most important practical result is the ability to use Equation (18) to determine the high frequency response for a transforming cell of arbitrary geometry. The calculation requires the solution of only the stationary diffusion equation and the equations of hydrodynamics. The result should be the local values of the components of the concentration $\frac{\partial {\tilde{c}}_{st}\left(\tilde{\xi },0\right)}{\partial \tilde{z}}$ and $\frac{\partial {\tilde{c}}_{st}\left(\tilde{\xi },0\right)}{\partial \tilde{\xi }}$ and the functions τ(ξ), n(ξ), characterizing the distribution of the hydrodynamic velocity in the vicinity of the cathodes. Then, the output current is calculated from Equation (18).

Moreover, Equation (18) can be used to optimize the geometry of the transforming element. For example, the geometry of the channels for the flow of the working fluid can be set, then find the areas on the channel walls with the maximum values of *τ*(ξ), *n*(ξ), place the cathodes in the vicinity of these areas, and then place the anodes in such a way as to obtain the maximum value of the integral in the Equation (18).

However, increasing the conversion factor cannot be the only goal of optimizing the design of the converting element. A more important criterion for assessing the ability to measure extremely small signals is the comparison of the signal-to-noise ratio for different designs of the conversion element. The approaches developed in this work can also be generalized for modeling the self-noise of a converter in the high-frequency region if we consider the hydrodynamic velocity as a random variable, the spatial and temporal correlation of the values of which can be found by solving the equations of hydrodynamics with Langevin random sources in the right-hand side similar to the works [27,36,37]. The authors suggest that the use of this approach may be a topic for further research.

## Figures and Tables

**Figure 1 micromachines-13-00153-f001:**
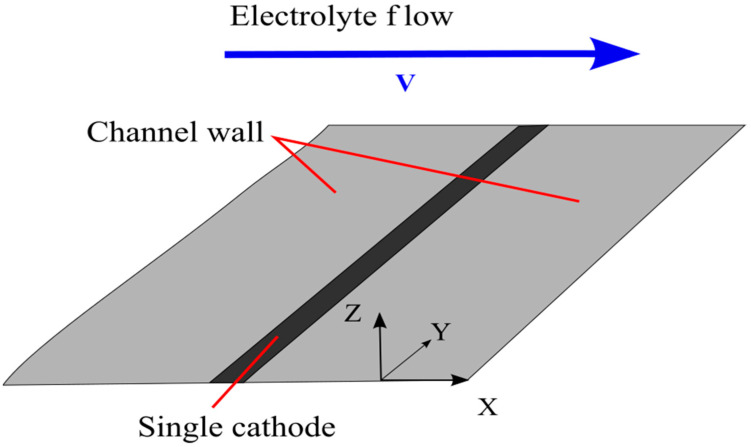
Single planar electrode.

**Figure 2 micromachines-13-00153-f002:**
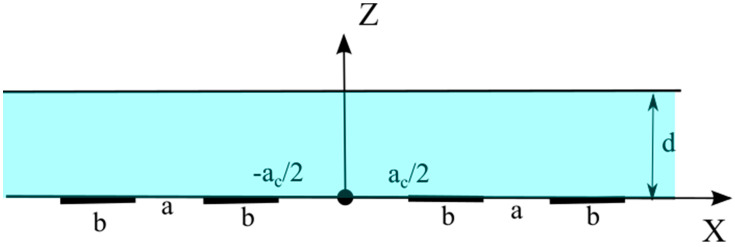
Schematic of the planar electrochemical cell used for stationary concentration calculation.

**Figure 3 micromachines-13-00153-f003:**
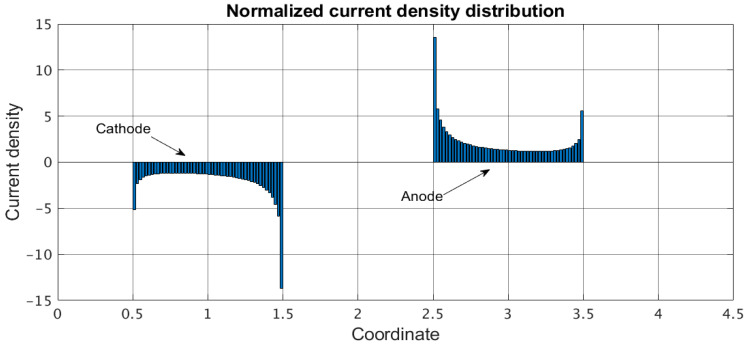
Dimensionless current density distribution. Transforming cell parameters: $\tilde{d}=2,\;\tilde{a}=1,\;{\tilde{a}}_{c}=1$. Only portions of electrodes corresponding to the positive coordinates are shown.

**Figure 4 micromachines-13-00153-f004:**
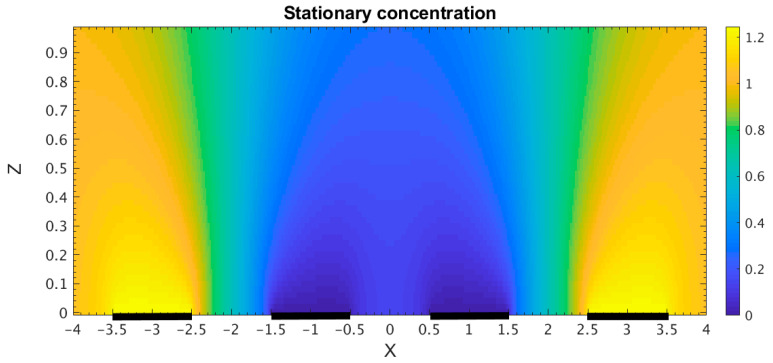
Dimensionless stationary concentration in the electrochemical cell vs. coordinates (shown by color). Thick black lines correspond to the electrodes position.

**Figure 5 micromachines-13-00153-f005:**
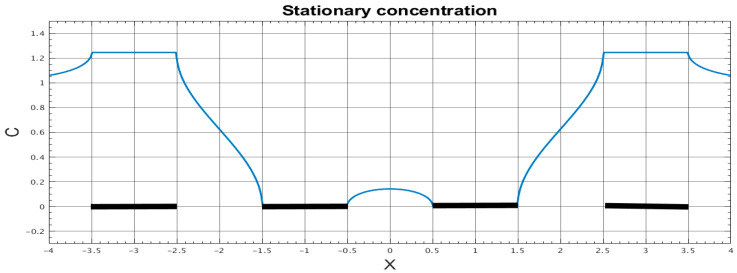
Dimensionless stationary concentration vs. coordinate on the electrochemical cell channel walls. Thick black lines correspond to the electrodes position.

**Figure 6 micromachines-13-00153-f006:**
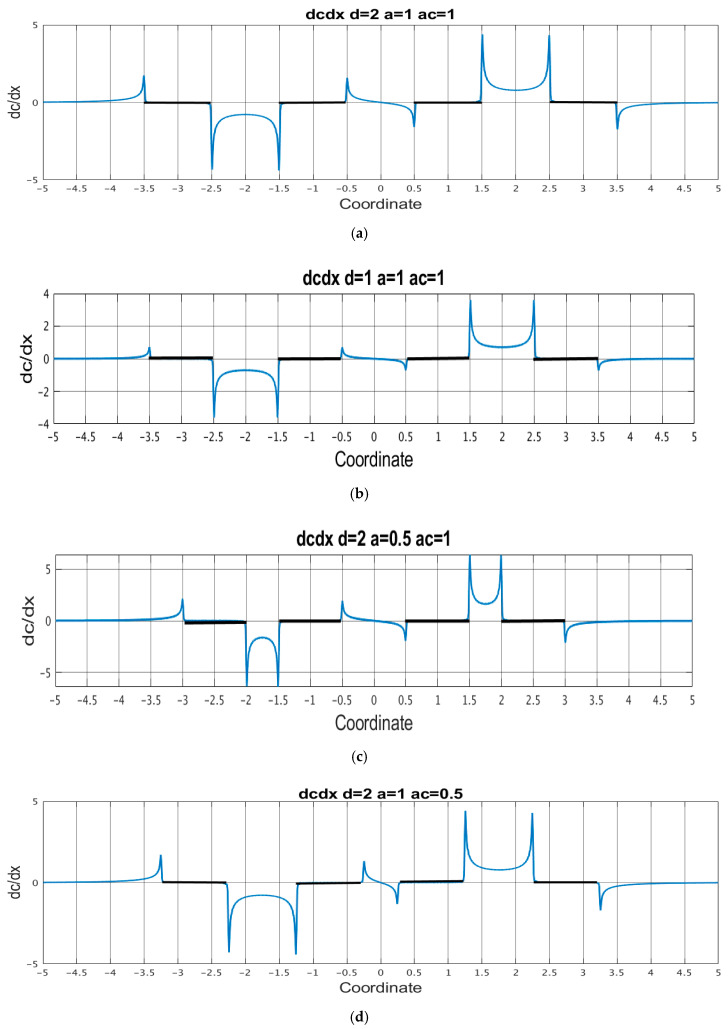
Tangential component of the stationary concentration gradient distribution. (**a**) $\tilde{d}=2;\tilde{a}=1;\;{\tilde{a}}_{c}=1;\left(b\right)\tilde{\;d}=1;\tilde{a}=1;\;{\tilde{a}}_{c}=1$; (**c**) $\tilde{d}=2;\tilde{a}=0.5;\;{\tilde{a}}_{c}=1;$ (**d**) $\tilde{d}=2;\tilde{a}=1;\;{\tilde{a}}_{c}=0.5$. The black thick lines show the areas on the channel wall where the electrodes are located.

**Figure 7 micromachines-13-00153-f007:**
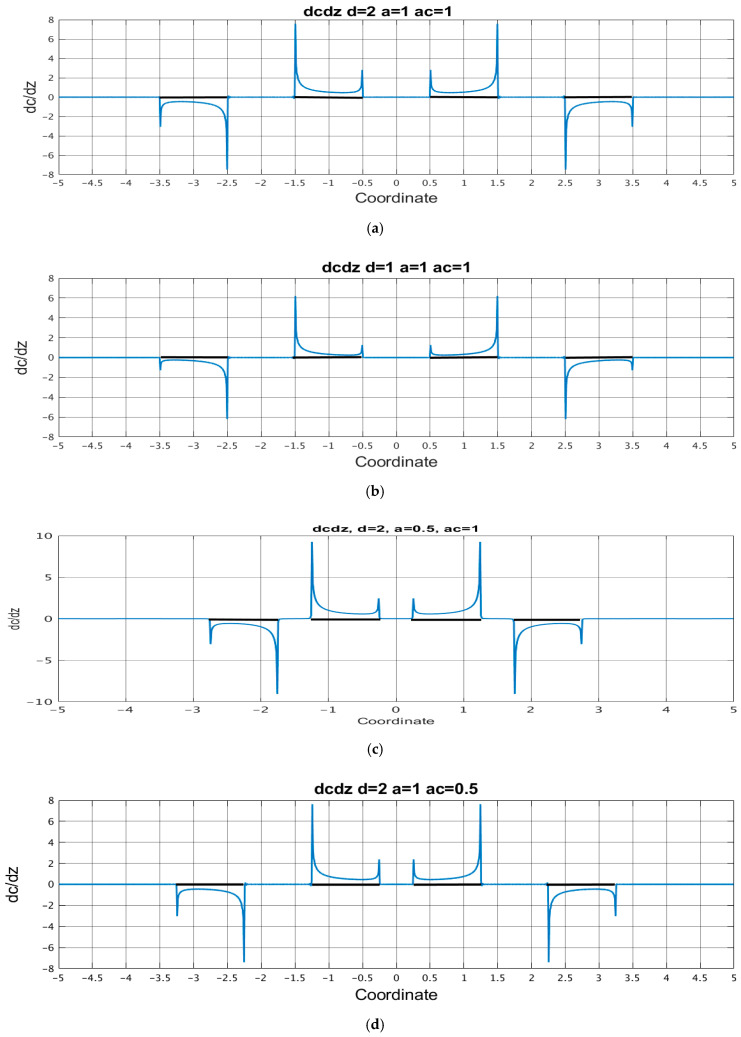
Normal component of the stationary concentration gradient distribution. (**a**) $\tilde{d}=2;\tilde{a}=1;\;{\tilde{a}}_{c}=1;\left(b\right)\tilde{\;d}=1;\tilde{a}=1;\;{\tilde{a}}_{c}=1$; (**c**) $\tilde{d}=2;\tilde{a}=0.5;\;{\tilde{a}}_{c}=1;$ (**d**) $\tilde{d}=2;\tilde{a}=1;\;{\tilde{a}}_{c}=0.5$. The black thick lines show the areas on the channel wall where the electrodes are located.

**Figure 8 micromachines-13-00153-f008:**
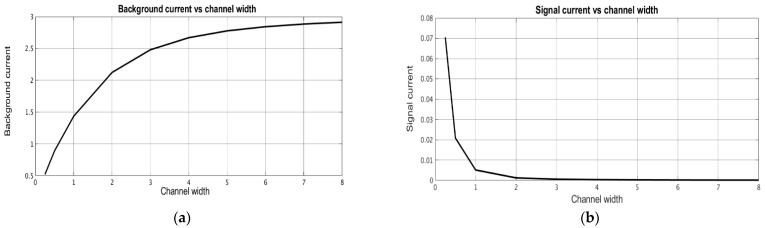
Dependence of (**a**) stationary background current and (**b**) signal current on dimensionless channel thickness.

**Figure 9 micromachines-13-00153-f009:**
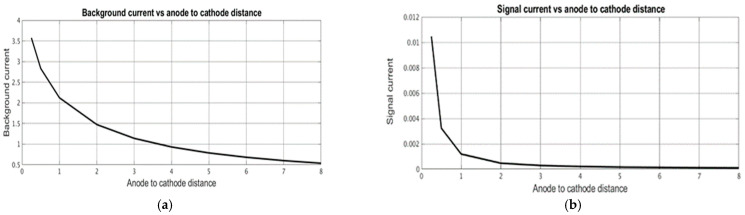
Dependence of (**a**) stationary background current and (**b**) signal current on the dimensionless distance between anode and cathode.

**Figure 10 micromachines-13-00153-f010:**
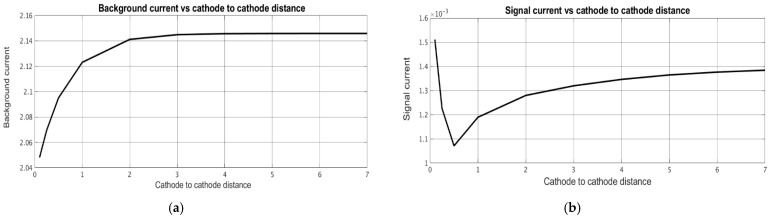
Dependence of (**a**) stationary background current and (**b**) signal current on the dimensionless distance between cathodes.

**Figure 11 micromachines-13-00153-f011:**
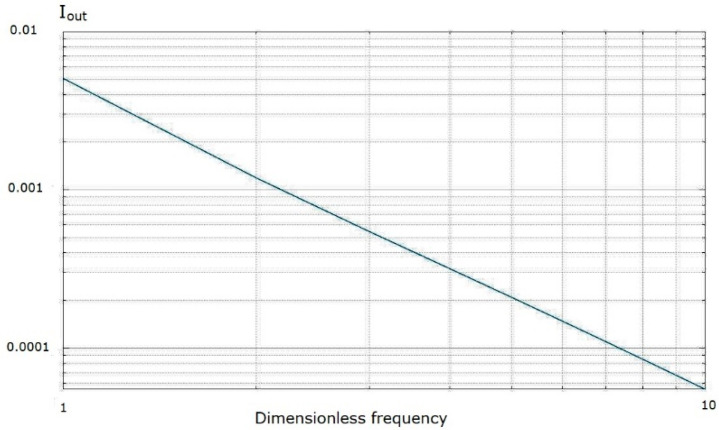
Output current frequency response.

**Table 1 micromachines-13-00153-t001:** Absolute values $\left|\frac{\partial {\tilde{c}}_{st}\left(\tilde{\xi },0\right)}{\partial \tilde{z}}\right|$ и $\left|\frac{\partial {\tilde{c}}_{st}\left(\tilde{\xi },0\right)}{\partial \tilde{\xi }}\right|$ near the cathode edges.

Cell Parameters	$\left|\frac{\partial {\tilde{c}}_{st}\left(\tilde{\xi },0\right)}{\partial \tilde{z}}\right|$, Adjacent Anode Side	$\left|\frac{\partial {\tilde{c}}_{st}\left(\tilde{\xi },0\right)}{\partial \tilde{z}}\right|$, Cathode Side	$\left|\frac{\partial {\tilde{c}}_{st}\left(\tilde{\xi },0\right)}{\partial \tilde{\xi }}\right|$, Adjacent Anode Side	$\left|\frac{\partial {\tilde{c}}_{st}\left(\tilde{\xi },0\right)}{\partial \tilde{\xi }}\right|$, Cathode Side
$\tilde{d}=2;\tilde{a}=1;\;{\tilde{a}}_{c}=1;$	7.58 ± 0.08	2.81 ± 0.03	4.39 ± 0.04	1.59 ± 0.02
$\tilde{d}=1;\tilde{a}=1;\;{\tilde{a}}_{c}=1$;	6.21 ± 0.06	1.26 ± 0.02	3.68 ± 0.03	0.7 ± 0.01
$\tilde{d}=2;\tilde{a}=0.5;\;{\tilde{a}}_{c}=1;$	11.2 ± 0.09	3.20 ± 0.02	6.42 ± 0.06	1.94 ± 0.02
$\tilde{d}=2;\tilde{a}=1;\;{\tilde{a}}_{c}=0.5$	7.63 ± 0.08	2.38 ± 0.02	4.42 ± 0.04	1.33 ± 0.02

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
