# Peer review of "Modeling of the Electrochemical Motion Sensor Conversion Factor at High Frequencies"

_micromachines, 2022, doi:10.3390/mi13020153_

Round 1

Reviewer 1 Report

In this work, the authors develop a theoretical model allowing to establish the basic laws that govern the conversion of high-frequency signals in an electrochemical cell.

In detail, the authors try to understand how to solve the problem that at high frequency, the conversion coefficient of electrochemical sensors decreases in a sensible way and the signal conversion is really challenging since it depends on the microscopic structure of the converting system.

The main finding is that the high-frequency conversion factor increases more by decreasing the inter-electrode distance and the channel width with respect to the increase obtained by increasing the cathode dimension. Therefore, an increase in the conversion coefficient can be achieved by increasing the size of the electrodes and, more effectively, by decreasing both the distance between the electrodes and the thickness of the channel.

The work is well written, and also the underlying mathematics is well presented. The results are sound, and no serious faults occur throughout the manuscript. Only minor corrections should be performed before the paper can be accepted for publication:

  • English language sometimes is not appropriate; already in the abstract some sentences should be rephrased (e.g. lines 25-27)
  • 29 is not visible
  • In figure 3 axes labels are missing
  • Line 252 “which confirms” should read “which confirm”
  • Captions of figures 4 and 5 should be revised.
  • Which is the error of the values presented in table 1?
  • Relevant literature on this topic should be added
  • Practical implications should be more profoundly discussed

Finally, for the exposed reasons, I believe the manuscript can be accepted for publication on Micromachines after performing the suggested corrections.

Author Response

Authors thanks reviewer for the attention and analysis. The answers to reviewer comments are given in the attached file.

Reviewer 2 Report

In the introduction, the author describes the case of high frequencies "when the diffusion length becomes less than the distance between the electrodes, it is sufficient to consider the transformation of the fluid motion only near one of the electrodes (cathode).". It seems not enough about the scientific problem between the diffusion length and electrodes distance. It is suggested of increasing the background of what about the low frequencis.

Author Response

Authors thanks reviewer for the comments. The detailed answers is given in the attached file.

Reviewer 3 Report

This manuscript develops a theoretical model that allows to establish the basic laws governing the conversion of high-frequency signals in an electrochemical cell. Overall, this manuscript is well organized and informative, I have several concerns for the authors’ consideration:

Major concern:

  1. I notice that the authors have published another paper entitled “Modeling of the MET Sensitive Element Conversion Factor on the Intercathode Distance” in Sensor (Sensors 2020 Sep; 20(18): 5146.). In that paper, the authors discussed the behavior of the conversion coefficient depending on the frequency. My question is the innovation of the current manuscript compared to the published one?

  1. The authors found that hat high-frequency conversion factor increases with inter-electrode distance and channel width decreases in comparison with the cathode dimension. My question is that what’s the importance of the conversion factor? In addition, are there any experimental results that can support this conclusion?

  1. What’s the relationship between the sensitivity of electrochemical sensors and conversion efficiency?

Minor concern:

  1. The schematic illustration of the planar electrode in Figure 1 is confusing.
  2. It is not clear that why choose molecular iodine as the study model.

  1. Not sure the meaning of the caption of Figure 2.

  1. There are several label issues in the Figures, for example, what is the label for x-axis/y-axis in Figure 3, Figure 4, and Figure 5?

There is no (a), (b), (c), or (d) mark in Figures 5, 6, 7, and 8 to indicate which one is which.

In addition, I notice that there are many equations in this manuscript that might cause a burden and distraction for the readers. As the authors have published a similar paper, I would like to suggest presenting these equations in a smarter way.

Author Response

Authors thanks reviewer for accurate reading and for very useful comments and suggestions. Point by point answers are given in the attached file.

Round 2

Reviewer 3 Report

I think the revised version is ready to publish.